# In-Line Near-Infrared Spectroscopy Gives Rapid and Precise Assessment of Product Quality and Reveals Unknown Sources of Variation—A Case Study from Commercial Cheese Production

**DOI:** 10.3390/foods12051026

**Published:** 2023-02-28

**Authors:** Lars Erik Solberg, Jens Petter Wold, Katinka Dankel, Jorun Øyaas, Ingrid Måge

**Affiliations:** 1Nofima—Norwegian Institute for Food, Fisheries and Aquaculture Research, Muninbakken 9-13, Breivika, 9291 Tromsø, Norway; 2TINE SA, Bedriftsveien 7, 0950 Oslo, Norway

**Keywords:** in-line NIR, process analysis, cheese production, power spectral density, sampling regime

## Abstract

Quality testing in the food industry is usually performed by manual sampling and at/off-line laboratory analysis, which is labor intensive, time consuming, and may suffer from sampling bias. For many quality attributes such as fat, water and protein, in-line near-infrared spectroscopy (NIRS) is a viable alternative to grab sampling. The aim of this paper is to document some of the benefits of in-line measurements at the industrial scale, including higher precision of batch estimates and improved process understanding. Specifically, we show how the decomposition of continuous measurements in the frequency domain, using power spectral density (PSD), may give a useful view of the process and serve as a diagnostic tool. The results are based on a case regarding the large-scale production of Gouda-type cheese, where in-line NIRS was implemented to replace traditional laboratory measurements. In conclusion, the PSD of in-line NIR predictions revealed unknown sources of variation in the process that could not have been discovered using grab sampling. PSD also gave the dairy more reliable data on key quality attributes, and laid the foundation for future improvements.

## 1. Introduction

Food products are routinely analyzed for a variety of reasons, including to comply with regulatory and labeling requirements, to meet product specifications, to ensure food safety, and for research, development, and innovation purposes. Quality testing is therefore an integral part of food production. 

Grab sampling means taking a small portion of a particular *lot* (e.g., a batch or production cycle) to the laboratory for testing. The sample is assumed to be representative of the whole lot. Different sampling regimes such as random, stratified, systematic or composite sampling can be applied to minimize the sampling error [1,2]. Nevertheless, grab sampling of heterogeneous food products is challenging and often leads to substantial sampling biases. Other challenges with grab sampling are time and resources. The time delay between sampling and analysis results limits the possibility for using the measurements in real-time process monitoring and control systems.

In-line measurement systems are viable alternatives to grab sampling for many quality attributes. Sensors placed directly in the process stream, either over a conveyor belt or in/onto a pipe or processing tank can automatically measure a large proportion of the lot, in real time, and therefore overcome many of the challenges of grab sampling.

Near-infrared spectroscopy (NIRS) is a technology that is based on illuminating a sample and analyzing the amount of transmitted or reflected light. This information is used to measure the properties of a sample. It is well known that NIRS is suitable for the in-line measurement of a range of food quality attributes related to chemical composition. The technology is widely used in the food industry, typically for the estimation of fat, water, protein, and carbohydrates in products such as meat, fish, cereals, and fruits [3]. Recent reported examples are industrial monitoring of dry matter in large batches of potatoes [4] and in-line quality sorting of chicken fillets [5]. NIR spectroscopy is a well-established method for the rapid determination of fat, protein, and moisture in cheese [6], and NIRS is widely used in the dairy industry [7]. Several studies have shown that dry matter and fat can be measured in cheese in the laboratory [8,9,10], while in a modern dairy there is a need to measure these features in-line on large cheese blocks weighing e.g., 20 kg. Eskildsen et al. [11] showed that it is possible to estimate the average dry matter and fat content of blocks of Swiss cheese with satisfactory accuracy by using in-line scanning NIRS in reflectance mode. Good results were obtained on cheese blocks both before and after pressing. This is possible when the surface chemistry of the cheese is representative of the interior.

Continuous measurements of quality attributes of food products define time series and different tools for analyzing the process variation in more detail becomes available. One such tool is the power spectral density (PSD), which provides a decomposition of signal power (variation) in the frequency domain and is particularly useful when periodic components exist. Sometimes frequency components may be related to distinct causes. In this article, we used the PSD on predictions from NIR spectra.

There are numerous examples of this. For instance, in [12], the authors studied the flow of particles suspended in a gas through a pipe using electrical capacitance tomography at two imaging planes. They used PSD to analyze aspects of the flow and argued that the probable cause of one peak in the PSD was due to a slip/stick condition at the particle inlet into the pipe. They also linked a second peak to periodic variations in the radial distribution of the particle flow.

In [13], experiments were carried out to relate the information in the PSD to process parameters in a laboratory-scale wet mill. They first divided the frequency domain into three bands and for each showed probable causes: the low-frequency band was linked to the mill, the mid-frequency linked to balls–mill shell lining interactions and the high-frequency range linked to ball–ball interactions. Further, via their experimental design, they related this information to parameters defining the operation of the mill which was subsequently used to formulate soft sensors. Similar uses of the power spectral density can be found in [14], where it was used to distinguish between possible physical processes in liquid-solid pipe flows; in [15] it was used as one part of a system for diagnosing stator faults in induction motors; and in [16] where three signal sources for diagnostics were compared using the PSD.

To the best of the authors’ knowledge, no publication has so far used the power spectral density as a diagnostic tool in the food industry.

In [17], a review of the role of NIRS for food quality analysis shows a list of achievements including but not limited to: milk and yoghurt attributes in the dairy industry; monitoring fermentation processes in wine and beer production; the drying of fruit and vegetables post-harvest; and dry ingredient mixing. However, the authors concluded that much had been performed to demonstrate the feasibility of applications in laboratory conditions but identify a gap regarding actual implementations at the industrial scale.

The aim of this paper is therefore to document some of the benefits of in-line measurements at the industrial scale, including the advantages over traditional grab sampling whilst also showcasing the possibilities of using PSD for continuous in-line measurements as a diagnostic tool. We achieve this by presenting an actual case study from cheese production, where in-line NIRS was implemented to replace traditional lab measurements of dry matter in Gouda-type cheese. In doing so, we will focus more on the temporal dimension of the approach rather than conducting an in-depth analysis of the spectral information.

## 2. Materials and Methods

This paper is based on data from a large cheese dairy that produces several types of semi-hard cheeses. Cheese making is a complex biochemical process, and its industrial production involves a series of highly controlled processing steps: a vat is filled with pasteurized milk with a standardized protein/fat ratio; starter culture and rennet are added thereby coagulating the milk and forming a curd; the curd is then cut and heated inside the vat, after which most of the whey is drained off. The remaining curd/whey mixture is transferred to a “casomatic” (Tetra Pak^®^ Tebel, Netherlands), which consists of vertical columns where the curd particles fuse together to form solid cheese at the bottom. Cheese blocks are then sliced off as they exit the column and are transported on a conveyor belt to a press. They are then placed in a brine bath and stored in a controlled environment until they reach the desired maturity. Figure 1 shows a schematic overview of the process with emphasis on the steps that are most relevant in this work.

Among the most important quality attributes of semi-hard cheeses is the dry matter content as it affects ripening and eating quality as well as profitability. Even if the process is highly standardized and controlled, the dry matter may vary substantially from batch to batch due to uncontrollable variations in raw milk quality, starter culture and rennet.

### 2.1. Data Collection

Data were collected during two time periods: (i) during test installations of an in-line NIRS system in 2019 and (ii) after full implementation of the same system in 2022. The dairy produces several different cheese types, but this study is limited to the one with the largest production volume.

In May–July 2019, a test installation of the NIRS instrument was run for a period of 18 production days at control point 1, CP1, and 11 production days at CP2 (see Figure 1). The same instrument was used at both time points. This test included data from the production of (i) 36,794 cheese blocks from 404 cheese vat batches at CP1, and (ii) 22,435 cheese blocks from 245 cheese vat batches at CP2.

In 2022, a permanent implementation of the same NIRS system was installed at CP1. About one week of continuous measurement from the permanent installation was included in this work, corresponding to 8327 cheese blocks from 94 batches.

### 2.2. Lab Measurements of Dry Matter

Grab samples were manually collected at a the control point directly after the brine bath, denoted as CP3 in Figure 1. Samples were routinely taken from approximately every fourth batch, by cutting a cross-section of 2 cm diameter from the center of one cheese block. According to the dairy’s protocol, samples were taken from a cheese block in the middle of the batch to account for a within-batch gradient caused by the casomatic (see Section 2.4). The dry matter was measured on a benchtop NIR system (FoodScan^TM^, FOSS Analytics, Hillerød, Denmark) using commercial calibration models.

### 2.3. In-Line Measurements of Dry Matter

The NIR spectrometer (DA 7440, Perten Instruments, Hägersten, Sweden), designed for in-line measurements on conveyor belts, was placed approximately 25 cm above the moving cheeses at either control points CP1 or CP2 (see Figure 1). These points were selected based on previously conducted trials in the dairy’s pilot plant [11]. The cheese blocks were transported in plastic containers with each block weighing approximately 20 kg. Spectra in the wavelength range 950–1650 nm were acquired continuously with a time resolution of 4 spectra/second. Spectra were aggregated to a single spectrum per cheese and pre-processed using the standard normal variate (SNV) approach.

Partial Least Squares (PLS) calibration models for dry matter were developed individually for CP1 and CP2, with both models calibrated to predict dry matter at CP3. This meant estimates from all three control points were directly comparable, even if the actual dry matter varied between them. Average calibration spectra at CP1 and CP2 are shown in Figure 2 with systematic differences between the two control points (which lead to slightly different models).

For the test installation in 2019, provisional calibration models based on approximately 50 and 30 samples were developed for CP1 and CP2, respectively. Samples were split in an 80–20% ratio between training and test sets. A 10-fold cross validation was used on the training set to select the number of components. The models obtained RMSEP values of 0.63% and 0.38% dry matter at CP1 and CP2, respectively, based on calibration samples spanning approximately 2% dry matter. These models are not very accurate as their RPD values were 1.3 and 2.3, respectively, but they still capture relative differences between cheeses that allow for greater understanding of some sources of variation in the process. In 2022, the calibration model for CP1 was significantly improved by the instrument vendor, by adding more samples and refining the model. The updated model spanned a range of 3.5% dry matter and obtained a RMSECV of 0.36%.

### 2.4. Data Handling and Integration

During the test installation in 2019, the spectrometer was not integrated in the dairy’s Manufacturing Execution System (MES) and a three-step procedure was therefore developed to assign single spectra to cheese blocks and further to the corresponding cheese vat batch:Aggregating spectra for each cheese block: The instrument was set up to provide an indication of valid spectra (corresponding to “cheese” and not background) based on a prior model. The temporal separation between two cheeses was used to aggregate the sequence of spectra into a single spectrum per cheese. This association of spectra to cheeses is not perfect, and errors occur leading to a variation in the length of batches in terms of the number of cheeses. It will be shown later that this complicates the decomposition.Assigning cheese blocks to batches: This was accomplished based on the shape of the time series of dry matter estimates; the casomatic (Figure 1) induced a gradient in the dry matter content in the resulting cheeses. This effect of increasing dry matter within the batches was well known and is illustrated in Figure 3. Transitions between batches were found by considering the expected negative jump in dry matter values by approximately 1%. This was discovered using an appropriate smooth derivative filter in the time domain and by looking for local maxima at temporal intervals approximately 90 cheeses.Linking the identified batches to MES: The batches from step 1 were matched with the MES system based on time stamps and the approximate time delays between the cheese vat and CP1/CP2. Data on other production parameters such as temperature, pH and durations of different processing steps in the cheese vat could then be exported from the MES and laboratory databases and then combined with the in-line measurements.

All steps were performed in a semi-automated way, meaning that the automatically obtained results were manually checked and corrected if necessary. Nonetheless, the procedure is prone to error, particularly in determining the transition between batches in step 2. This is illustrated in Figure 3 (observe e.g., the transition between light blue and dark red batches at approximately 10:15).

In the full implementation in 2022, the spectrometer was integrated with the MES to ensure full traceability between single cheese blocks and cheese vat batches. However, missing values were detected whenever the cheese ID was not a contiguous sequence, or a new batch did not start with cheese number 1, or the length of the batch was not an integer number of 4 cheeses (the number of columns in the casomatic). These missing values have been imputed in an attempt to make the series as consistent as possible with respect to the casomatic column. The imputation was based on the known triangular form of the casomatic contribution and performed separately for each column: $x_{i}=a\cdot i+b$, where a,b were estimated from the cheeses in the same batch and from the same column.

### 2.5. Time Series Trend Analysis

The in-line NIRS measurements formed a time series, and it was already known that the dry matter content of a single cheese was given by the cheese vat batch mean as well as its order in the casomatic step (within a batch), which gave the cyclic trends shown in Figure 3. It was therefore natural to decompose the dry matter variation into components representing the batch mean, a linear casomatic trend and residuals.

The production ran continuously for periods, with pauses due to cleaning or production of other cheese types in between. These pauses were used to split the entire data set into subsets of continuous time series, denoted “sequences”. The time series analyses were performed separately on each sequence, and were assumed to be continuous.

#### 2.5.1. Decomposition of the Time Series

The dry matter time series signal (x[n]) was decomposed by first considering the overall mean (*μ*). The centered sequence (x[n]−μ) was then split into batches where least squares models including a constant and linear term were fit to each batch separately. The batch offsets ($x_{b}\left[n\right]$) were equated to the constant terms while the casomatic contributions ($x_{c}\left[n\right]$) were associated to the linear components. However, due to the uncertainty related to batch transitions, only the central 80% of the samples were used for fitting the model. The remaining signal was denoted the residual ($x_{r}\left[n\right]$). The full decomposition was therefore expressed as:(1)$$x\left[n\right]=\mu +x_{b}\left[n\right]+x_{c}\left[n\right]+x_{r}\left[n\right].$$

Note, the batch mean was the sum of the first two terms: $\mu +x_{b}\left[n\right]$.

#### 2.5.2. Power Spectral Density

Stationary processes such as this can be described using their power spectral density (PSD), which is the Fourier transform of a signal’s autocorrelation function:(2)$$R_{X}\left(\tau \right)=E\left[x\left(t\right)x\left(t-\tau \right)\right]$$
(3)$$S_{X}\left(f\right)=ℱ\left(R_{x}\left(\tau \right)\right)$$
where $R_{X}$ is the autocorrelation function of the signal x, τ the time lag, ℱ the Fourier transform and $S_{X}$ the PSD as a function of frequency f. Equation (3) leads to the relationship between the power of the signal and the PSD [18]:(4)$$E\left[x^{2}\left(t\right)\right]=R_{x}\left(0\right)=\int^{\;}S_{X}\left(f\right)df$$
which implies that the power associated with any frequency interval can be found by integrating the PSD over that interval. The $S_{X}\left(f\right)$ is hence the expected power per unit frequency. For instance, the casomatic signal with a period of say 92 cheeses would have a peak in its PSD spectrum at a (fundamental) frequency of $f_{1}=1/92$, along with components at multiples (k) of the fundamental frequency $f_{k}=kf_{1}$. This set (sometimes with the exception of the fundamental) is collectively referred to as the harmonics. This is a property shared by all periodic signals and can be used to identify other periodic sources of influence on the process. A perfectly periodic signal has non-zero PSD values only at its harmonic frequencies, while signals with small variations in period or shape will give non-zero PSD values over regions around these frequencies.

The implementation of the PSD is based on the discrete signal (sampled in time) and using Welch’s method by averaging the estimates of different windows of the signal within a sequence [19]:(5)$${\hat{S}}_{X,N}^{\left(i\right)}\left(f\right)=\frac{{\left|X_{N}^{\left(i\right)}\left(2\pi f\right)\right|}^{2}}{U\cdot N},\;\;f\in \left[-1/2,\;1/2\;\right]$$
where i indexes the window used for the distinct estimates ${\hat{S}}_{X,N}^{\left(i\right)}$ of the PSD based on individual, overlapping windows of length N and $X_{N}^{\left(i\right)}$ is the discrete time Fourier transform (DTFT) of the time series $w_{N}\left[n\right]\cdot x_{N}^{\left(i\right)}\left[n\right]$. The restriction on f is due to sampling and has units of cycles/sample. Also, to increase contrast in the PSD the *Hann* window ($w_{N}$) was used with appropriate scaling ($U={\left\|w_{N}\right\|}^{2}/N$) of the PSD estimates.

Estimates of time series components corresponding to quasi-periodic signals can be obtained based on filtering in the frequency domain with narrow passbands around its harmonic frequencies. It is worth noting that this procedure reliably separates components only when the different components have a disjoint support in the frequency domain, i.e., the peaks in the PSD do not overlap.

### 2.6. Comparison of Sampling Regimes by Simulation

The main aim of the dry matter measurements was to estimate the batch mean, and in-line NIR was expected to give more precise estimates than the traditional grab sampling regime. The estimates may have been affected by cyclic trends in the time series, and a simulation study was set up to compare different sampling regimes for various magnitudes of cyclic trend components. Our results showed that there are two cyclic trends present at CP2, caused by the casomatic and press steps (described in Section 3.1). Data were therefore simulated to resemble the observed dry matter time series at CP2, since this control point was assumed to be similar to CP3, where the grab sampling was performed. A continuous time series of N = 920,000 cheese blocks (10,000 batches with 92 units in each batch) was constructed by summing independent components representing batch mean ($x_{b}$), casomatic contribution ($x_{c}$), press contribution ($x_{p}$) and noise ($x_{e}$):(6)$$x\left[n\right]=\mu +x_{b}\left[n\right]+x_{c}\left[n\right]+x_{p}\left[n\right]+x_{e}\left[n\right]$$

The batch mean $x_{b}$ was constant over all 92 units in a batch. The casomatic effect $x_{c}$ was simulated as a linear cyclic effect with positive slope and of period 92, and the press effect $x_{p}$ as a linear cyclic effect with negative slope and period 30. The noise was independent and normally distributed. 

Four different time series were simulated, with varying size of the casomatic and press components:-*Simulation 1*: the press component was set to zero, mimicking a case where there was only one cyclic trend which was aligned with the batch.-*Simulation 2*: the size of the press and casomatic components were similar to the observed data, i.e., the variation of the casomatic component was approximately 3.5 times the press component (see Table 2).-*Simulation 3*: the magnitude of the press component was set to be 3.5 times the casomatic component, to mimic a situation where the dominating cyclic trend was not aligned with batches.-*Simulation 4*: the casomatic component was set to zero, mimicking a case where there was only one cyclic trend which was not aligned with the batch.

For each of these time series, four different sampling regimes were performed:*Grab random*: the batch mean was represented by a randomly selected unit.*Grab mid precise*: the batch mean was represented by one of the 2 middle units in a batch.*Grab mid sloppy*: the batch mean was represented by one of the 20 middle units.*In-line*: the batch mean was estimated as the average over all units in a batch.

The precision of the batch mean estimates were evaluated by mean squared error (MSE) and the coefficient of determination (R^2^). The variations in estimated batch means were also calculated, for comparison with observed data. The four simulated data sets are provided in the Appendix A.

## 3. Results

### 3.1. Time Series Trend Analysis

#### 3.1.1. Initial Decomposition of the Time Series

The time series of dry matter estimates was initially decomposed into contributions representing the grand mean, batch offset, casomatic, and residuals, as described in Equation (1). The power spectral density (PSD) for each of these components are given in Figure 4 for the 2019 data and in Figure 5 for the data from 2022. In these figures, the sequence mean (μ) has been removed from the signal for clarity. When comparing the 2019 and 2022 power spectral densities, it was worth keeping in mind that the former relied on many more batches and sequences for estimating the same PSD and hence appeared less noisy. This was especially apparent in the grey residual levels. In the following, we will primarily discuss the 2019 data, and add comments regarding the 2022 data where appropriate.

The batch offsets represented a low-frequency signal quickly decreasing in power density with increasing frequency: this was as expected because within each batch, this signal was constant. The casomatic component had clearly visible peaks at approximately 1/92 and its harmonics, both at CP1 and CP2. This was also as expected since each batch consists of approximately 92 cheese blocks. The residuals were assumed to contain only measurement noise and no autocorrelation or periodic trends. If so, the PSD should have been flat with only random fluctuations. This was however not the case for either of the control points, indicating that there was more structured variation in the signal.

At CP1, the residuals showed a peak around the frequency f=1/4, thus with a period of 4 cheeses. The casomatic had four parallel columns, and it was likely that the peak came from a systematic difference between these. The reconstruction of this signal in the time domain was not stable: the contribution of each of the four columns changed rapidly from batch to batch. While the PSD strongly suggested a component at $f_{0}=1/4$, a likely reason for the poor reconstruction was the errors associated with the assignment of spectra to individual cheeses, as described in Section 2.4. The random sequence of “duplicated” or “missing” cheeses effectively mixed the contributions of columns when these were taken as every fourth cheese. It was also not clear if the sequence of cheeses from the four columns necessarily repeated the same order or if the true sequence may have been somewhat random. The variance of this component was only significant in the CP1 period: it would appear that the press effectively equalized the contribution from different columns, or else that the press re-organized cheeses so that the different columns no longer corresponded to every fourth cheese.

The same remarks held approximately for the 2022 data with a couple of comments. First, the batch offset signal was stronger and had a softer decay with frequency. Inspection of the time series showed that most sequences had large jumps where the dry matter estimate either rose or fell abruptly—and more so than for the 2019 data. Such jumps would have tended to increase the high-frequency content in the signal. Secondly, the peaks associated with the casomatic appeared to be slightly wider. It is not clear what may have been causing this, but it was possible that this may have been related to how batch lengths vary.

At CP2, the residuals had clear peaks at a fundamental frequency of approximately 1/30 along with its harmonics. Given that the only processing step between CP1 and CP2 was the press, and that this press had three arms each containing 30 cheeses, this variation was very likely induced by the press.

#### 3.1.2. Refined Decomposition of the Time Series

The newly discovered components coming from the columns (at CP1) and press (at CP2) were extracted from the residuals using narrow bandpass filters, and updated residuals were calculated by subtracting the reconstructed signal. A new PSD was then calculated on the refined decomposition, see Figure 6 for the 2019 data and Figure 7 for the 2022 data. Again, for clarity, the sequence mean has been removed as in Figure 4. The updated residuals showed only weak peaks in the PSD. In the CP1 period, this approach to decomposing the columns signal effectively created a dip in the PSD at 0.25 and showed that this component’s estimate included a part of an uncorrelated noise component. There were also elevated power regions at approximately 0.13 and 0.18 cycles/sample; these have not been associated with a stage in the industrial process. However, their strength was fairly low as they were only barely above the noise floor. Finally, in the CP2 period there was a peak at approximately 0.2 cycles/sample which was in fact the 6th harmonic of the press signal—the filtering had been limited to include the 5th harmonic.

Regarding the 2022 data for CP1, the tracking of cheeses by the Manufacturing Execution System avoided many of the errors induced by the procedure for identifying individual cheeses for the 2019 data even if some missing data had to be imputed. One consequence was that the signal associated with the difference between columns was significantly more concentrated around the ¼ cycles per sample frequency. However, if the total power of this source of variation were to remain the same, this peak should have been equivalently stronger. This was not observed. To the contrary, the peak level was slightly lower and power was therefore reduced, as confirmed in Table 1.

The reconstruction of the signals in the time domain can be observed in Figure 8, together with the single-cheese estimates. For more clarity, instead of displaying each signal separately, the successive approximations to the dry matter signal are shown by accumulating the components. One important observation is that the press signal at CP2 had a period of approximately 30 cheeses, which was not a multiple of the casomatic signal period of 92 cheeses. This meant that the press signal was not aligned with the batch, implying that the fixed grab sampling of cheeses at the middle of a batch would experience the full variation of the press signal.

The average power (variance) of the different components is provided in Table 1. This shows that batch offsets contributed the most to variance while the casomatic and residuals accounted for a large part of the remaining variance.

The variance of the casomatic ($\sigma_{c}^{2}$), given its shape as a linear increase within each batch, may be directly related to the amplitude (A) of the difference between first and last cheese within the batch: $\sigma_{c}^{2}=A^{2}/12$. This implied an average amplitude at CP1 of 0.99 and at CP2 of 0.90. On the other hand, the batch offsets showed higher variance at CP2 than in CP1. Two factors may have contributed to this: (i) measurements did not correspond to the same period, as for the casomatic, and (ii) the high variability of the batch offset’s contribution was such that the threshold of significant differences increased.

The approximation to the dry matter variation signal including the press component, shown in Figure 8, compensated for the slope resulting from the casomatic signal: within each sequence of 30 cheeses, the trend was either flat or falling where it had been increasing without the press signal. One could speculate that the effect of the press was to equalize the dry matter among the cheeses in the same arm, either through a difference in pressure on each cheese, or it may have also been easier to increase dry matter for cheeses with more liquid.

With regard to the 2022 data, the observations above also apply with an adjustment to the various components’ strengths, as shown in the third row of Table 1. The main difference was due to a significant increase in the power of the batch offset component. As mentioned earlier, there were significant jumps in this signal, which explained the increased high-frequency content. However, the scale of these jumps also explained the added power: whatever caused these jumps were major sources of variation in the batch offset signal. The second difference related to the casomatic signal, which had slightly increased power mainly due to an increase in the average amplitude of this signal.

As noted earlier, a decomposition in the frequency domain was reliable when the components had disjoint supports. This was not the case for the casomatic and press signals: the 3rd, 6th, 9th, etc., harmonics of the casomatic substantially overlapped with the press signal. In our decomposition, the extraction of the casomatic signal was based on the assumption of a linear dependency which limited the influence between the two components. This influence was further reduced due to a limited correlation following from a difference in periods.

### 3.2. Comparison of Grab Sampling and In-Line Measurements

The dairy wanted to minimize the batch-to-batch variation in dry matter, and a correct estimation of the process variation was therefore important. Table 2 shows the estimated process variation for different simulated data sets, along with variation in the real observed data. The in-line sampling (i.e., average over all units in a batch) always reproduced the correct variation, while all the grab sampling regimes over-estimated the process variation. Of the grab sampling regimes, better results were obtained by selecting a unit from the middle of the batch if cyclic trends were present. Note however that it was not important to select the unit that was exactly in the middle, as the difference between “sloppy” and “precise” mid sampling was negligible. It was interesting to note that the observed data showed comparable results to the simulation results. This indicated that the process variation was smaller than previously believed. In Figure 9, we zoomed into four production days that clearly showed that the grab samples varied more than the in-line measurements.

**Table 2 foods-12-01026-t002:** Variation in estimated batch means for different sampling regimes. The true variation is 0.15 for all simulated data sets.

	Grab Random	Grab Mid Sloppy	Grab Mid Precise	In-Line
**Simulation 1**	0.28	0.21	0.20	0.15
**Simulation 2** ^1^	0.28	0.22	0.22	0.15
**Simulation 3**	0.30	0.27	0.27	0.15
**Simulation 4**	0.27	0.28	0.27	0.15
**Real data (CP1)**		0.29 ^2^		0.09 ^3^
**Real data (CP2)**		0.28 ^2^		0.14 ^3^

^1^ Cyclic components have same size as observed data. ^2^ Based on data from lab database. ^3^ Based on in-line NIR.

Table 3 sums up the precision in estimation of batch mean, based on simulated data. As expected, the random grab sampling gave consistently poor results for all data sets, while the “grab mid” regimes gave slightly better results for data sets where the dominant cyclic trend was aligned with the batch. In-line sampling gave excellent precision in all cases as averaging over 92 units effectively reduced the error to almost zero, with non-zero values only in the fourth decimal of RMSE. Note however that prediction error of the NIR calibration model was not included here, so the precision (of all sampling regimes) was expected to be lower in the real application.

## 4. Discussion

In-line measurements have many advantages over traditional grab sampling, also beyond the case illustrated here. With regard to precision, the added value of in-line measurements is higher when the lot is more heterogeneous. Appropriately designed sampling regimes, such as stratified or composite sampling, may also give representative grab samples with precise estimates of within- and between-lot variations. However, such regimes are often labor intensive, and in-line NIRS will in many cases be a more viable method. It is also clear from the results section that the average over a batch in this case improved the precision of batch estimates in addition to reducing the impact of asynchronous, periodic processes (the press signal) as well as high-frequency contributions (e.g., the columns).

A higher temporal resolution greatly improves process understanding and makes it possible to assess the variation within and between e.g., batches, production days, raw material providers and operator shifts. Also, portable and robust devices that can be moved between different measurement points gives the opportunity to distinguish between sources of variation in different steps of the process. Such data and insights are important for identifying and prioritizing improvement areas, e.g., as part of *lean manufacturing, six sigma* or *sustainable production* methodologies [20]. The details that such continuous measurements provide may also improve the sensitivity of detecting faults or anomalous production conditions and the capacity to localize the source of such problems.

In the case study presented here, the use of in-line NIRS at two locations in the process has opened the door for a time-frequency and power spectral density analysis of the process. The differences observed along the production process have enabled an attribution of variation that would otherwise have been more difficult to document. This analysis has also identified components that were not previously known: the press’ and columns’ contributions to dry matter variation.

The success of a time-frequency analysis of the data in this case study has relied on the presence of periodic contributions as periodic signal components are very well distinguished: observe the spikes in Figure 6 as opposed to the time-domain signal in Figure 8 where the general slope is visible (casomatic) but other components are less so. This analysis allowed for the identification of a significant contribution from the casomatic based on the fundamental frequency and knowledge of the number of cheeses corresponding to a batch. For the same reasons, a second important contribution could be identified with a fundamental frequency of 30 cheeses, which in this case could be attributed to the press, both because of knowledge of the press (30 cheeses per arm), but also from comparing measurements at two locations. A third periodic component was also observed, and a probable association to the columns of the casomatic was suggested. Finally, it was possible to estimate a flat, background noise level. Such a flat spectrum is related to uncorrelated processes, meaning the part of the next dry matter measurement is uncorrelated to all other measurements. This noise level is best understood in terms of measurement uncertainty (or error) as it is unlikely that the process itself adds a unique, independent contribution to each cheese.

Having obtained the decomposition, partly based on prior knowledge and partly on observations in the time-frequency analysis, it was possible to quantify the sources of variation. This quantification showed the relative contributions of the components in the model, see Table 1. This quantification, along with knowledge of its variability may allow for more precise detection of when an anomaly is occurring, e.g., small changes in the casomatic would be difficult to detect in the total dry matter signal given that the batch offset and residuals variations are large.

A second important point relates to the casomatic signal. Increasing dry matter through a batch is directly related to the position of the corresponding block in the columns of the casomatic and this relationship is basically due to a physical phenomenon. Therefore, while the signal induced variation in dry matter, it basically added the same variation every time as its cause was stable. This can be observed in how little this variance differed between “sequences” (the parentheses in Table 1). At the other end, the estimate of variation due to batch offsets differed significantly from one “sequence” to the next: there were likely upstream causes that induced these differences. The fact that there were such differences suggested that its causes were unstable and may in part be open to optimization, after all there were sequences with low variance.

Having distinguished sources of variation, it may be desirable to reduce these. An obvious target for reduction would be the casomatic as it was found to be the processing step responsible for most of the within-batch variation. It might not be possible to remove this variation without changing the processing technology, but knowledge about it can give rise to mitigation strategies such as re-ordering of the cheese blocks in subsequent processing steps or sorting cheeses to different storage conditions. Another major contribution to variance is the batch offset. While its variance and instability has been quantified, further detective work would be necessary to relate this variation to possible causes, of which raw material variation and differences in process settings are usual suspects.

Among the advantages of time-frequency analysis mentioned above was the ability to distinguish signals that are periodic with different fundamental frequencies. Such signals are expressed as a series of spikes in the frequency domain, such as those observed in Figure 6. However, time-frequency analysis is not limited to this category of signals: many processes occupy wide bands in the frequency domain. When describing signals as “smooth”, this normally translates into signals with a limited support in the frequency domain: there is little power above a certain frequency. In other words, the “speed” of the process is related to the highest significant frequency component, and variation e.g., due to different work shifts (approximately changing every 8 h) would occupy a larger band than seasonal variations (months).

This paper describes the retrospective analysis of in-line measurements. An even bigger potential of in-line NIRS lies in the real-time operationalization of the process measurements, usually coupled with results from retrospective analysis. Such applications may for instance be anomaly/fault detection, real-time process adjustments and control, and the sorting of products into different quality categories.

## 5. Conclusions

Continuous in-line measurements of food quality have several advantages over grab sampling. Firstly, they may provide more precise estimates by reducing sampling bias. Secondly, time dynamics may be revealed by analyzing auto-and cross-correlation patterns. Thirdly, real-time measurements allow for rapid response to unwanted deviations and anomalies thus improving process control. This work documented the first two of these three advantages by analyzing data from a large-scale cheese production plant. We showed that the variations in dry matter as observed by grab sampling were larger than those measured in-line, implying that the process is more stable than previously believed. We also showed that the frequency domain decomposition revealed three different periodic components of variation in dry matter, stemming from different processing steps. The implementation of in-line NIRS therefore gave the dairy new insights into the process that could not have been discovered using grab sampling, and laid the foundation for future improvements.

## Figures and Tables

**Figure 1 foods-12-01026-f001:**
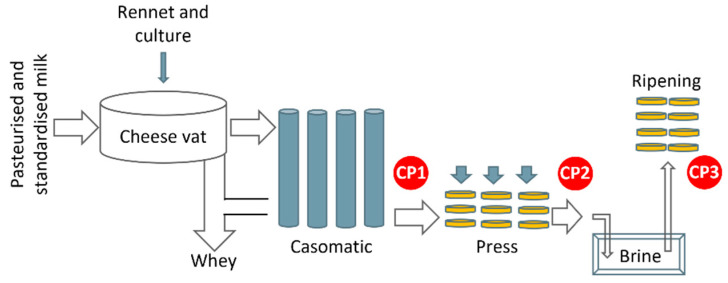
Schematic overview of the cheese production process. Control Points (CP) are marked with red circles. The in-line NIRS sensor was positioned at either CP1 or CP2, while the grab samples were taken at CP3.

**Figure 2 foods-12-01026-f002:**
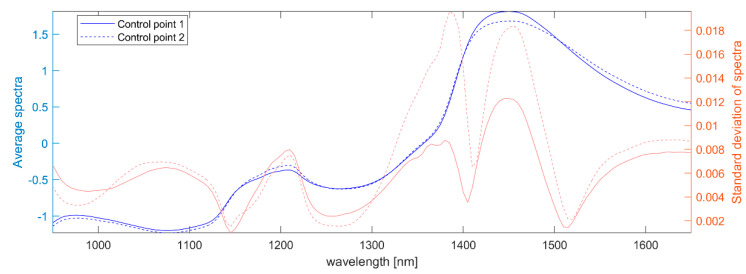
Standard normal variate (SNV) pre-processed calibration spectra, 1/log(X) transformed for absorbance, from the in-line NIRS instrument: mean (left axis, blue) for each of the control points (drawn with different line styles) along with standard deviations (right axis, orange).

**Figure 3 foods-12-01026-f003:**
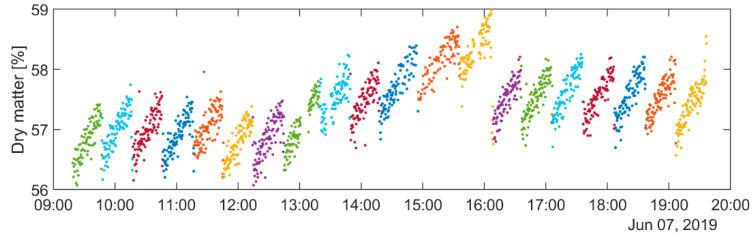
In-line estimates of dry matter at CP1 for a randomly selected production day in 2019. The colours indicate grouping into batches, identified based on the well-known gradient in dry matter within each batch caused by the casomatic.

**Figure 4 foods-12-01026-f004:**
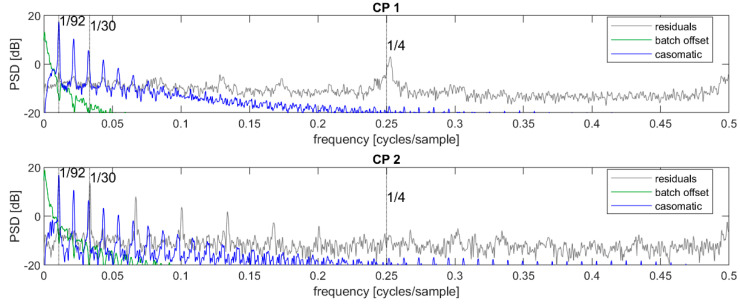
Top: power spectral density of initial decomposed signal at control point CP1; bottom: same for signal at CP2—both from 2019. The unit is in the logarithmic scale of decibel (dB) which implies a factor 10 for each increase by 10 dB.

**Figure 5 foods-12-01026-f005:**
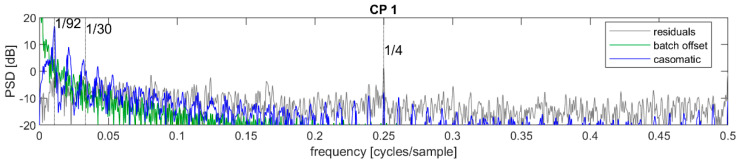
Power spectral density at CP1 for the 2022 data. The unit is in the logarithmic scale of decibel (dB) which implies a factor 10 for each increase by 10 dB.

**Figure 6 foods-12-01026-f006:**
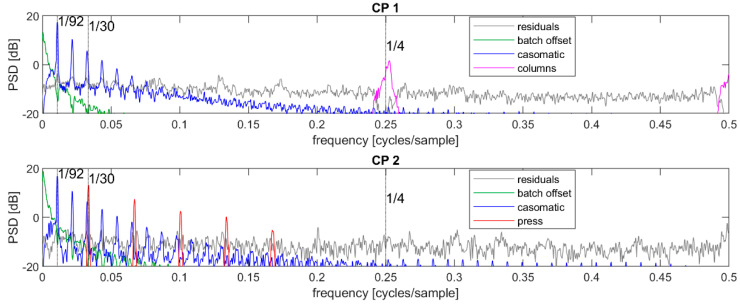
Top: power spectral density of the refined decomposed signal at control point CP1; bottom: same for signal at CP2—both from 2019. The unit is in the logarithmic scale of decibel (dB) which implies a factor 10 for each increase by 10 dB.

**Figure 7 foods-12-01026-f007:**
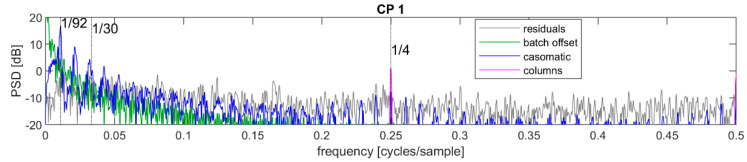
Power spectral density of the refined decomposed signal at control point CP1 for the 2022 data. The unit is in the logarithmic scale of decibel (dB) which implies a factor 10 for each increase by 10 dB.

**Figure 8 foods-12-01026-f008:**
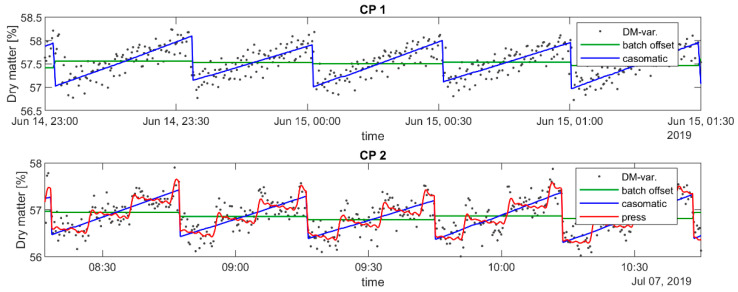
Zoom on a short period of the decomposition of dry matter measurements at CP1 and CP2—both from 2019. These decompositions are displayed as successive accumulations: $x_{\mu }+x_{b},\;x_{\mu }+x_{b}+x_{c}$ etc. The CP1 period does not include the press signal, and neither include the “column” ($x_{\delta }\left[n\right]$) signal.

**Figure 9 foods-12-01026-f009:**
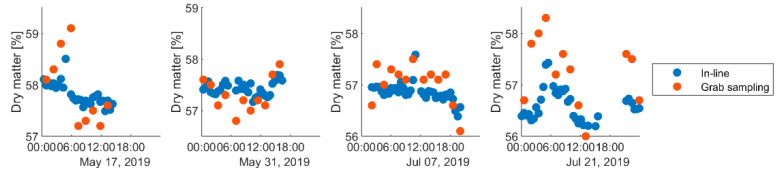
Estimates of batch means for four selected production days in 2019. Grab samples are taken from approximately every fourth batch. The estimates show the same trends, but variation is larger in grab samples.

**Table 1 foods-12-01026-t001:** Average (standard deviation) over sequences of variance of the dry matter signal and of its different components. An estimate of the “columns” contribution at CP2 is left out as the PSD suggested that this component is dominated by the noise level.

Control Point	Dry Matter (Variance) $\sigma^{2}$	Batch Offsets $\sigma_{b}^{2}$	Casomatic $\sigma_{c}^{2}$	Columns $\sigma_{\delta }^{2}$	Press $\sigma_{p}^{2}$	Residuals $\sigma_{r}^{2}$
CP1^2019^	0.20	0.092	0.082	0.0069		0.043
	(0.078)	(0.074)	(0.009)	(0.002)		(0.006)
CP2^2019^	0.28	0.15	0.068		0.017	0.048
	(0.11)	(0.093)	(0.010)		(0.005)	(0.007)
CP1^2022^	0.70	0.55	0.12	0.0023		0.047
	(0.26)	(0.25)	(0.024)	(0.0004)		(0.017)

**Table 3 foods-12-01026-t003:** Precision of batch mean estimates for different sampling regimes, evaluated by mean squared error (MSE) and R^2^.

	Grab Random		Grab Mid Sloppy		Grab Mid Precise		In-Line	
	**MSE**	**R^2^**	**MSE**	**R^2^**	**MSE**	**R^2^**	**MSE**	**R^2^**
**Simulation 1**	0.14	0.52	0.05	0.74	0.05	0.76	0.00	1.00
**Simulation 2** ^1^	0.13	0.53	0.07	0.68	0.07	0.68	0.00	1.00
**Simulation 3**	0.14	0.53	0.12	0.56	0.12	0.56	0.00	1.00
**Simulation 4**	0.12	0.55	0.12	0.56	0.12	0.56	0.00	1.00

^1^ Cyclic components have same size as observed data.

## Data Availability

Data are proprietary to TINE SA and have not been made publicly available.

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
