# Peer review of "In-Line Near-Infrared Spectroscopy Gives Rapid and Precise Assessment of Product Quality and Reveals Unknown Sources of Variation—A Case Study from Commercial Cheese Production"

_foods, 2023, doi:10.3390/foods12051026_

Round 1

Reviewer 1 Report

The manuscript ‘In-line near infrared spectroscopy gives rapid and precise assessment of product quality and reveals unknown sources of variation – a case study from commercial cheese production’ aims in investigating the benefits of NIR spectroscopy for the rapid quality control in food industry. Authors proved that such technique provides a better estimate of the dry matter during cheese production with respect to traditional grab sampling. In addition, PSD was used as a tool for the better understanding of the process. In fact, it gives a general overview of the process and could also act as a diagnostic tool. The manuscript is well written, great scientific soundness and has a worth to be published with minor revision.

Some specific comments:

- the selection of the keywords could be improved;

- Line 32: Is there already any standards in this sector that ensures a representative sampling?

-          Line 47-57: the state of art could be improved adding more bibliography.

-          Line 58-62: please explain better

-          Fig. 1: please improve the resolution of the figure

-          Fig. 1: why are the samples taken from C3? Are there any other control points at the moment so to check the production before the final product?

-          Line 139: is there already a technical standard for ensuring the collection of a representative sample and the performance of the following lab analysis?

-          Line 150: did the authors try to avoid averaging the spectra? And why is SNV chosen as pretreatment?

-          Lines 152-157: are the PLS models be adjusted to predict CP3 samples working on data collected on CP1 and CP2?

-          Lines 158-165: some information about the PLS models is missing; e.g. R2, type of validation.

-          Line 275-278: maybe these sentences could be moved to the results section?

-          Figure 8: please add the year in the figure caption.

-          Table 1: please add the year in table 1.

-          Line 469: why not adding this information to the supplementary material?

 General comment:

-          I was wondering if authors investigated the spectral differences of cheese samples between the three different arms of the press and the four columns of the casomatic.

-          Why have the authors not considered applying ASCA?

Author Response

Dear reviewer

Thank you for having invested time in a thorough review of our paper. You have provided a number of points that we will address below so our answer will be easier to relate to your remarks.

Best regards,

Lars Erik Solberg (on behalf of the authors)

==================================================================================

The manuscript ‘In-line near infrared spectroscopy gives rapid and precise assessment of product quality and reveals unknown sources of variation – a case study from commercial cheese production’ aims in investigating the benefits of NIR spectroscopy for the rapid quality control in food industry. Authors proved that such technique provides a better estimate of the dry matter during cheese production with respect to traditional grab sampling. In addition, PSD was used as a tool for the better understanding of the process. In fact, it gives a general overview of the process and could also act as a diagnostic tool. The manuscript is well written, great scientific soundness and has a worth to be published with minor revision.

Some specific comments:

  • the selection of the keywords could be improved;

We are not sure which keywords the reviewer is thinking about but agree that at least one is missing that should cover the sampling aspect: we have added “sampling regime” to keywords.

  • Line 32: Is there already any standards in this sector that ensures a representative sampling?

In this case, there is no common or regulatory defined standard for sampling of product quality. When we refer to a “protocol”, this is something the dairy producer has defined.

  • Line 47-57: the state of art could be improved adding more bibliography.

We have provided 9 specific references on NIR spectroscopy on foods and on cheese specifically, as well as at least one review article. Our focus in this article is not so much a review as going through an example application. We therefore hope that the reviewer will accept that we do not add more references.

  • Line 58-62: please explain better

The use of the power spectral density is more thoroughly explained later on in the article but we acknowledge that the brief account in these lines is not sufficient. We have tried to improve on this paragraph by adding some information on why the frequency domain is relevant.

  • 1: please improve the resolution of the figure

A higher-resolution image has been created.

  • 1: why are the samples taken from C3? Are there any other control points at the moment so to check the production before the final product?

The control point CP3 belongs to the industrial line’s standard quality procedures and was not determined by this study – as opposed to CP1 and CP2.

  • Line 139: is there already a technical standard for ensuring the collection of a representative sample and the performance of the following lab analysis?

See answer to line 32.

  • Line 150: did the authors try to avoid averaging the spectra? And why is SNV chosen as pretreatment?

Our approach here aimed at providing one measurement per cheese, hence the aggregation over all spectra which were obtained on the same cheese.

We have attempted both median and averaging without observing much of a difference. Selecting a single (best?) sample was not tested as it was considered risky, nor were other aggregation methods attempted.

Regarding SNV, it is a simple and robust technique that often provides as good results as many other methods. We have attempted some others without any significant improvements.

  • Lines 152-157: are the PLS models be adjusted to predict CP3 samples working on data collected on CP1 and CP2?

Yes. We hope this is clear enough as the text currently reads.

  • Lines 158-165: some information about the PLS models is missing; e.g. R2, type of validation.

Thank you, we have added some notes on this in the article. We have chosen RPD over R2 and hope this is ok for the reviewer – both describe the quality of the model but RPD has an easier interpretation in the opinion of the authors.

  • Line 275-278: maybe these sentences could be moved to the results section?

While this does use results before being presented in the results section, its use here is to justify how the simulation study has been constructed (and hence addresses methods and materials) and not to show or discuss results.

  • Figure 8: please add the year in the figure caption.

Thank you, we have added the year of data acquisition to all figures (where relevant).

  • Table 1: please add the year in table 1.

This information is already present in the caption for table 1, but perhaps in a somewhat less evident way. We have adjusted the table to make this more evident.

  • Line 469: why not adding this information to the supplementary material?

We have removed the comment “(not shown)”. We hope the reviewer agrees that this information is not key to understanding the article.

General comment:

  • I was wondering if authors investigated the spectral differences of cheese samples between the three different arms of the press and the four columns of the casomatic.

In relation to studying the spectral differences between the three arms of the press and the four columns of the casomatic, we have not investigated this. It is a good question, or proposal, as it opens up a new door into understanding the process. However, while spectral analysis is a necessary part of an in-depth study of applied NIRS, we choose to limit the scope of the article to the time dimension – the “inline” part of the study. This is not clearly stated in the article, and we have therefore added a note about scope in the introduction (last line).

  • Why have the authors not considered applying ASCA?

ASCA is a multivariate extension of ANOVA. It is suitable for analyzing effects of factors on a multivariate response when data is collected according to an experimental design. It is not clear to the authors how the present data can fit into an ASCA approach.

Reviewer 2 Report

1. For introduction

Please added more information about NIRs and PSD, how NIRs works, and what is the relationship between NIRs and PSD.

2. For English changes 

Moderate English changes required throughout the manuscript.

Author Response

Dear reviewer

Thank you for having invested time in a thorough review of our paper.

With respect to “Please added more information about NIRs and PSD, how NIRs works, and what is the relationship between NIRs and PSD.”, we have attempted to address this in the introduction, in particular in lines 45-47 (NIR) and 60-66 (PSD and relation to NIR). Given that the special issue is all about NIR in industrial applications, and that we present the PSD more in detail in the methods and materials section, we hope the small changes adequately address your concerns.

Regarding the “Moderate English changes required throughout the manuscript.” we have considered your specific suggestions as well as a new review of the article. We have made multiple changes (hopefully improvements) throughout. We hope the language is now adequate for the journal.

Best regards,

Lars Erik Solberg (on behalf of the authors)